# NEW TRAINING FRAMEWORK FOR SPEECH ENHANCEMENT USING REAL NOISY SPEECH

## ABSTRACT

Recently, deep learning-based speech enhancement (SE) models have gained significant improvements. However, the success is mainly based on using synthetic training data created by adding clean speech with noise. On the other hand, in spite of its large amount, real noisy speech is hard to be applied for SE model training because of lack of its clean reference. In this paper, we propose a novel method to utilize real noisy speech for SE model training based on a non-intrusive speech quality prediction model. The SE model is trained through the guide of the quality prediction model. We also find that a speech quality predictor with better accuracy may not necessarily be an appropriate teacher to guide the SE model. In addition, we show that if the quality prediction model is **adversarially robust**, then the prediction model itself can also be served as a SE model by modifying the input noisy speech through gradient backpropagation. Objective experiment results show that, under the same SE model structure, the proposed new training method trained on a large amount of real noisy speech can outperform the conventional supervised model trained on synthetic noisy speech. Lastly, the two training methods can be combined to utilize both benefits of **synthetic** noisy speech (easy to learn) and **real** noisy speech (large amount) to form semi-supervised learning which can further boost the performance both objectively and subjectively. The code will be released after publication.

## 1 INTRODUCTION

Deep learning-based speech enhancement (SE) has gained significant improvements in different aspects such as model structures (Xu et al., 2014; Weninger et al., 2015; Fu et al., 2017; Luo & Mesgarani, 2018; Dang et al., 2022; Hu et al., 2020), input features (Williamson et al., 2015; Fu et al., 2018b; Huang et al., 2022; Hung et al., 2022), and loss functions (Pascual et al., 2017; Fu et al., 2018b; Martin-Donas et al., 2018; Kolbæk et al., 2018; Koizumi et al., 2017; Niu et al., 2020). However, the success is mainly based on synthetic training data, which includes different clean and noisy speech pairs. In general, the noisy speech is synthesized by adding clean speech with noise; hence, both clean speech and noise are required for model training. Compared to real noisy speech, pure clean speech and noise are very difficult to obtain in daily life, and they have to be recorded in a controlled environment. Although some studies (Wisdom et al., 2020; Fujimura et al., 2021) have been proposed to use real noisy speech for SE training, they still rely on synthetic training data by adding noise to noisy speech to generate a noisier signal as model input with the original noisy speech as the training target. The mismatch between synthetic training data and real noisy data may degrade the SE performance (e.g., the recording devices and the room responses of noisy speech and noise may be different, which results in different acoustic characteristics).

This study aims to solve the above-mentioned issues by training a SE model directly on real noisy speech. To achieve this goal, we first train a non-intrusive speech quality predictor. If this predictor is **robust**, then it should be able to guide the training of a SE model. Because the quality assessment can be done without the need for a clean reference, real noisy speech can be applied for SE model training. A few key characteristics of the proposed method are: 1) The training of the SE model is based on real noisy speech and a quality prediction model; no synthetic training data is required. 2) The loss function to train the SE model is not based on the signal level comparison (such as mean square error between the enhanced and target speech); it is completely based on the quality predictor.

To summarize the key contributions of the paper:

1) A novel training framework for speech enhancement using real noisy speech is proposed.

2) We found that a speech quality predictor with better prediction accuracy may not lead to a better SE model. **Model structure does matter!**

3) Adversarially robust quality predictor itself can directly be used for speech enhancement without the need to train a separate SE model.

4) Under the same SE model structure, the proposed new training method can outperform the conventional supervised trained model.

5) The conventional supervised training and proposed methods can be combined together to form semi-supervised learning and further boost the performance.

## 2 RELATED WORK

Previous research has proposed using real noisy speech for SE model training. It can be further divided into two categories depending on if clean speech or noise is needed.

**SE training using unpaired noisy and clean speech:** Cycle-consistent generative adversarial network (CycleGAN) (Xiang & Bao, 2020; Yu et al., 2021) is applied to achieve this goal. Through the framework of a GAN and cycle-consistent loss, only non-paired clean and noisy speech was needed during training. Bie et al. (2021) used clean speech to first pre-train a variational auto-encoder and applied variational expectation-maximization to fine-tune the encoder part during inference.

**SE training using noisy speech and noise signal:** MixIT (Wisdom et al., 2020) is an unsupervised sound separation method, which requires only mixtures during training. It can also be used in SE with some simple modifications. The input to the SE model is noisy speech and noise-only audio. A three-output SE model is trained; outputs 1 and 3 or 1 and 2 can be used to reconstruct the noisy speech, while outputs 2 or 3 can be used to match the noise-only audio. However, it was found that the performance is poor if the distributions between noise in the noisy speech and the artificially added noise are too different (Saito et al., 2021; Maciejewski et al., 2021). Trinh & Braun (2021) apply two additional loss terms based on Wav2vec 2.0 (Baevski et al., 2020) to improve the MixIT performance.

Similar to the input of MixIT, Fujimura et al. (Fujimura et al., 2021) proposed noisy-target training (NyTT) by adding noise to noisy speech. The noise-added signal and original noisy speech are used as the model input and target, respectively.

Compared to these methods, our model does not need a 'pure' noise or clean corpus but requires a data set with a MOS label. In addition, the loss function of our SE model is to maximize the predicted quality score, which may make the enhanced speech have higher subjective scores.

**SE with a quality predictor:** MetricGAN (Fu et al., 2019b; 2021) applies a GAN framework to make the discriminator mimic the behavior of perceptual evaluation of speech quality (PESQ) (Rix et al., 2001) function. Then the discriminator is used to guide the learning of the SE model by maximizing the predicted score. Xu et al. (2022) propose a non-intrusive PESQNet as the discriminator.

'NOn-matching REference based Speech Quality Assessment' (NORESQA) is proposed in (Manocha et al., 2021) to estimate the quality differences between an input speech and a non-matching reference. Then the authors apply the NORESQA model to pre-train a SE model by minimizing the predicted quality differences between the output of a SE model and a clean recording. Manocha et al. (2020) propose a perceptual distance metric based on just-noticeable difference (JND) labels, and the model is applied as a perceptual loss for SE training. In Nayem & Williamson (2021), joint training is applied to train a SE model together with a MOS predictor.

Because the calculation of PESQ and training of NORESQA rely on two signal processing measures, Signal-to-Noise Ratio (SNR) and Scale-Invariant Signal to Distortion Ratio (SI-SDR) to compare the quality of the two inputs, synthetic data is needed to train the quality prediction model. However, in our proposed training method, it is not necessarily needed.

# 3 METHOD

## 3.1 MOTIVATION

DeepDream (Mordvintsev et al., 2015) is a way to visualize what features, a particular layer of a classifier model have learned. To maximize the activations of that layer, gradient ascent is applied on the input image to modify its content. The resulting image will generally become psychedelic-looking with some visual cues (e.g., dog, and cat, etc.) of the classes that the classifier was trained on. Although it can not generate realistic image, this algorithm shows that the classifier can be used to generate corresponding features.

Another example is the discriminator in GAN (Goodfellow et al., 2014), which can also guide the generator to generate more realistic data. In addition, without the need of a generator, Santurkar et al. (2019) show that a single adversarially robust classifier can be applied to different kinds of image synthesis tasks by modifying the input image. The authors claims that adversarial robustness is precisely what we need to directly manipulate salient features of the input. Recently, Ganz & Elad (2021) use this technique as a post-processing step to further refine the generated images from different generative models and obtain SOTA results.

The above examples show that a classifier may not only give us a classified result but if we use it in an inverse way by modifying the input, the input may be changed to what we specified. Therefore, in this study, we explore whether we can get enhanced speech through a non-intrusive speech quality predictor such that real noisy speech can also be used for training (note that, here we train our quality predictor as a regression model instead of a classifier).

## 3.2 PROPOSED METHOD

The key to being able to use real noisy speech as training data for SE is based on a non-intrusive (no clean reference is needed) speech quality predictor. Depending on whether its weights will also be updated during SE model training and the requirement of a separate SE model, we propose three training schemes and a semi-supervised training method.

### 3.2.1 DEEPDREAM-LIKE TRAINING

Similar to the framework in DeepDream, we first train a speech quality predictor $Q$ using Eq. (1) and once the training converges, its weights are permanently fixed.

$$L_Q = \min_Q \mathbb{E}_x [Q(x) - MOS_x]^2 \tag{1}$$

where $x$ and $MOS_x$ are the training pair that comes from a corpus containing both the speech signal and its corresponding MOS label.

The quality predictor is then concatenated after a randomly-initialized SE model. To train the SE model, the weights in the SE model are updated to maximize the predicted quality score using Eq. (2).

$$L_{SE} = \min_{SE} \mathbb{E}_z [Q(SE(z)) - MOS_{max}]^2 \tag{2}$$

where $z$ can be **real** noisy speech (which may come from another training corpus), $SE$ is the SE model, and $MOS_{max}$ is the highest quality score. Note that here $Q$ is fixed and only $SE$ will be updated. Compared to DeepDream, there are two differences: 1) In our training method, the particular layer that we try to maximize its activation is the output layer. 2) In addition to the quality predictor, we also employ another model $SE$ to do speech enhancement.

### 3.2.2 GAN-LIKE TRAINING

Another training scheme follows the training style of GAN (Fu et al., 2022; Ravanelli et al., 2021), so both the speech quality predictor (similar to the role of discriminator in GAN) $D$ and the SE model

---

**Algorithm 1** Proposed adversarial attacks for speech enhancement

---

**Input:** noisy spectrogram $\mathbf{Z}$, quality prediction model $Q$, target score $S$
**Output:** enhanced spectrogram $\mathbf{Y}$
1: $\mathbf{noise} \leftarrow 0$
2: **for** $i \leftarrow 1$ to $I$ **do**
3:     $\mathbf{Y} \leftarrow \mathbf{Z} - \mathbf{noise}.\text{clip}(\text{min=0})$
4:     $\mathbf{g}_{noise} \leftarrow \nabla_{noise} L(Q(\mathbf{Y}), S)$
5:     $\mathbf{noise} \leftarrow \mathbf{noise} - lr \cdot \mathbf{g}_{noise}$
6:     $\mathbf{noise} \leftarrow \mathbf{noise}.\text{clip}(\text{min=0})$
7: **end for**
8: $\mathbf{Y} \leftarrow \mathbf{Z} - \mathbf{noise}.\text{clip}(\text{min=0})$

---

are alternatively updated. Therefore the loss function of the quality predictor is modified from Eq. (1) to Eq. (3):

$$L_D = \min_D \mathbb{E}_x [D(x) - MOS_x]^2 + [D(SE(x)) - MOS_{SE(x)}]^2 \tag{3}$$

Note that to obtain $MOS_{SE(x)}$, a human-in-the-loop (Zanzotto, 2019) is needed which is time and cost consuming. Therefore, in this study, we apply an off-the-shelf speech quality predictor to provide $MOS_x$ and $MOS_{SE(x)}$ in GAN-like training (see Section 4.1.5 for more detail).

### 3.2.3 SPEECH ENHANCEMENT BY A ROBUST QUALITY PREDICTION MODEL

Motivated from (Santurkar et al., 2019), we investigate whether an adversarially robust quality predictor itself can directly be used for speech enhancement without the need to train a separate SE model. Adversarial robustness is a property that a model will not change its prediction when applying small adversarial (in order to fool it) perturbation on a model's input. To obtain such a classifier $C$, one can solve the following optimization problem (Madry et al., 2017):

$$\min_C \mathbb{E}_x [\max_\delta L(C(x + \delta), Label_x)] \tag{4}$$

where $L$ is the loss function. We can approximate the solution of the inner maximization via adversarial attacks (Madry et al., 2017). Then, this optimization problem can be solved iteratively by fixing $C$ to optimize the perturbation $\delta$, and then fixing $\delta$ to update $C$. This training algorithm is also called adversarial training. As stated in (Engstrom et al., 2019; Ganz & Elad, 2021), after adversarial training, when modifying the input to maximize the target label, the gradient will become perceptually aligned gradients (PAG), such that the modified input will be perceptually aligned to the target label.

However, we believe Eq. 4 is mainly suitable for getting a robust classifier instead of a regression model. Because given a perturbation $\delta$, $x + \delta$ may still belong to the label $Label_x$ for a classification problem. On the other hand, in our regression case, $x + \delta$ may not match to $MOS_x$ anymore, i.e., the target label should be $MOS_{x+\delta}$. Therefore, we can apply our proposed Algorithm 1 for adversarial attacks and using a similar loss function as in Eq. 3 for adversarial training. The two clip functions (line 3 and 6) in this algorithm are used to prevent enhanced magnitude spectrogram $\mathbf{Y}$ has T-F bins smaller than zero, and constraint estimated noise spectrogram $\mathbf{noise}$ to be larger than zero (since we assume it is additive noise), respectively. When this iterative training converges, we then get a robust quality prediction model $Q_r$.

Once we get $Q_r$, we can apply Algorithm 1 again to do speech enhancement. Note that in this algorithm, the only model we need is the quality predictor, no SE model is required. This training framework is actually very similar to GAN-like training, the main difference is it removes the reliance on a generator.

### 3.2.4 SEMI-SUPERVISED TRAINING

To utilize both benefits of synthetic clean/noisy speech pairs (easy to learn) and real noisy speech (large amount), conventional supervised training method and our proposed approach can be combined

into a semi-supervised learning. Thus the knowledge from the two sides can be incorporated and better results are expected. For GAN-like training (Section 3.2.2), we can simply replace the randomly-initialized SE model with the supervised trained model. For the method proposed in Section 3.2.3, we can just replace the noisy spectrogram **Z** in Algorithm 1 with the enhanced spectrogram generated by the supervised trained model. In this sense, Algorithm 1 works like a post-processing as in (Ganz & Elad, 2021).

## 4 EXPERIMENTS

### 4.1 EXPERIMENTS SETTINGS

#### 4.1.1 SPEECH ENHANCEMENT MODEL

The enhancement model used in the proposed training methods (Section 3.2.1 and 3.2.2) is a CNN-BLSTM (Zhao et al., 2018; Tan & Wang, 2018). The architecture of the CNN has four 2-D convolutional layers, each with kernel size=(9, 9) and number of filters=32 for the first three layers, while the last layer only uses 1 filter. For the BLSTM model, it consists of two bidirectional LSTM layers, each with 200 nodes, followed by two fully connected layers, each with 300 LeakyReLU nodes and 257 sigmoid nodes for mask estimation, respectively. When this mask (between 0 to 1) is multiplied with the noisy magnitude spectrogram, the noise components should be removed. In addition, as there is no clean reference during training, the model may aggressively remove noise which may also harm the speech component. To solve this issue, during training, we used a clamping mechanism so that all predicted mask values smaller than a threshold will be mapped to the threshold (Koizumi et al., 2018).

#### 4.1.2 DATASET FOR TRAINING SPEECH ENHANCEMENT MODEL

In the scenario of semi-supervised SE model training, only a limited number of synthetic (noisy, clean) pair data can be available, and there is a lot of real noisy speech without its corresponding clean reference. In this study, we used the publicly available synthetic VoiceBank-DEMAND (VBD) dataset (Valentini-Botinhao et al., 2016) as the source of (noisy, clean) pair data. The conventional supervised SE baselines are also trained on this dataset. The training sets (11,572 utterances) consisted of 28 speakers with four signal-to-noise ratios (SNRs) (15, 10, 5, and 0 dB). We randomly selected two speakers (p226 and p287) from this set to form a validation set (770 utterances).

The VoxCeleb2 (Vox2) (Chung et al., 2018) dev dataset is served as real noisy speech training data. It contains over 1 million utterances for 5,994 celebrities, extracted from videos uploaded to YouTube. Among them, we randomly select 7 speakers (id04344, id03220, id09272, id00012, id06698, id07497, and id05423) and 10 speakers (id00019, id00995, id01452, id03379, id04178, id05384, id06114, id07223, id08098, and id09109) to form the validation (1,001 utterances) and test set (1,615 utterances), respectively.

Another test set is from the 3rd DNS challenge (Reddy et al., 2021a) test set (600 utterances) to evaluate the generalization ability of different SE models on totally different acoustic conditions.

#### 4.1.3 EVALUATION OF SPEECH ENHANCEMENT

Because the two test sets mentioned above do not contain the corresponding clean reference, we apply non-intrusive metrics to evaluate the SE performance. DNSMOS P.835 (Reddy et al., 2022) is a widely used metric for such scenarios. It is a neural network-based quality estimation metrics that can be used to evaluate different deep noise suppression (DNS) methods based on MOS P.835 estimates (noa, 2003). Three scores are provided in this metric: speech quality (SIG), background noise quality (BAK), and the overall quality (OVRL) of the audio. We use OVRL as the training stop criteria if it reaches the maximum score on the validation set. Although we will use the information from DNSMOS p.808 (Reddy et al., 2021b) for SE model training (Section 4.1.5), DNSMOS p.808 and DNSMOS p.835 are actually different models (i.e., model structure and training data). Therefore DNSMOS p.835 is still a **fair** measure in this study (we still show the DNSMOS p.808 scores just as a reference). In addition, in the following, unless specified otherwise, DNSMOS refers to DNSMOS p.808.

### 4.1.4 NETWORK ARCHITECTURE FOR QUALITY PREDICTION

To investigate the impact of the model structure on the quality prediction and SE performance, we prepare three different model structures:

1) **CNN**: We use a similar model structure as the discriminator used in MetricGAN. It is a CNN with four two-dimensional (2-D) convolutional layers with 15 filters and a kernel size of (5, 5). To handle the variable-length input, a 2-D global average pooling layer was added such that the features could be fixed at 15 dimensions. Three fully connected layers were subsequently added, each with 50 and 10 LeakyReLU neurons, and 1 linear node for quality score estimation.

2) **BLSTM**: A similar model structure as the one proposed in QualityNet (Fu et al., 2018a). It consists of two bidirectional LSTM layers, each with 200 nodes. After BLSTM, three time-distributed linear layers were subsequently added, each with 50 and 10 LeakyReLU neurons, and 1 linear node. The global average pooling layer was finally applied to map the frame-wise scores to utterance-wise.

3) **CNN-BLSTM**: As reported in MOSNet (Lo et al., 2019), concatenating BLSTM after a CNN can obtain better MOS prediction accuracy. In this study, we also follow this idea to construct a CNN-BLSTM for quality estimation.

### 4.1.5 DATASET FOR TRAINING QUALITY PREDICTION MODEL

The IU Bloomington (IUB) corpus (Dong & Williamson, 2020) is used to train the quality predictor, $Q_{IUB}$. There are 36,000 speech utterances, each truncated between 3 to 6 seconds long, with a total length of around 45 hours. For validation and test set, we randomly select 1800 utterances for each of them. In this corpus, each utterance has its corresponding MOS. Because it adopted ITU-R BS.1534 (noa, 2014) for subjective testing, which resulted in a rating range of 0∼100 instead of 1∼5, we first linear normalized the scores to between 1∼5.

In addition to the above-mentioned corpus, we also try to simulate a larger data set based on the Vox2 training data with the corresponding MOS score given by the DNSMOS p.808 (Reddy et al., 2021b) model. The quality predictor is hence called, $Q_{Vox2+DNSMOS}$.

To make the training more consistent for GAN training (1 represents real, and 0 for fake), before training the quality predictors, we further normalize the MOS score from 1∼5 to 0∼1 by a $Sigmoid$ operation.

### 4.1.6 EVALUATION OF PREDICTION ACCURACY

Linear correlation coefficient (LCC) (Pearson, 1920), Spearman's rank correlation coefficient (SRCC) (Spearman, 1961) and mean square error (MSE) between the true quality scores and the predicted ones are used to measure the performance of speech quality prediction.

## 4.2 EXPERIMENTS RESULTS

### 4.2.1 IMPACT OF THE MODEL STRUCTURE ON QUALITY PREDICTION AND SPEECH ENHANCEMENT

In the first experiment, we want to explore the impact of the model structure of quality predictor $Q$ on the quality estimation accuracy and effects of training speech enhancement models.

In Table 1, three quality predictors with model structures introduced in Section 4.1.4 are trained on the IUB training data with the training target as the corresponding MOS scores. After the training has converged, the quality predictors are concatenated after a randomly-initialized SE model (CNN-BLSTM, as introduced in Section 4.1.1). To train the SE model, the weights in the quality predictors are fixed and only the weights in the SE model are updated to maximize the predicted quality score using Eq. (2). From the table, we can first observe that this SE training scheme can really remove noise (BAK gets improved compared to the one in the noisy condition) and improve the overall speech quality (although some speech components may be harmed). Note that during SE model training, **no** clean reference is needed, only noisy speech and quality predictor are required. This verifies the possibility of using DeepDream-like training to get a SE model.

Table 1: Performance comparison of different model structures of quality predictor, $Q_{IUB}$ on the speech quality prediction (**IUB test set**) and speech enhancement (Vox2 test set).

| prediction model $Q$ | Quality prediction training material: (IUB) | | | Speech enhancement training material: (Vox2 + $Q_{IUB}$) | | |
| --- | --- | --- | --- | --- | --- | --- |
| | LCC | SRCC | MSE | SIG | BAK | OVRL |
| Noisy | - | - | - | **4.47** | 3.40 | 3.50 |
| CNN | 0.8223 | 0.8373 | $4.38 \times 10^{-4}$ | 4.45 | **3.72** | **3.70** |
| BLSTM | **0.8341** | **0.8588** | $\mathbf{4.13 \times 10^{-4}}$ | 4.45 | 3.49 | 3.59 |
| CNN-BLSTM | 0.8317 | 0.8570 | $4.24 \times 10^{-4}$ | 4.45 | 3.47 | 3.57 |

Table 2: Performance comparison of different model structures of quality predictor, $Q_{Vox2+DNSMOS}$ on the speech quality prediction (**Vox2 test set**) and speech enhancement (Vox2 test set).

| prediction model $Q$ | Quality prediction training material: (Vox2 + DNSMOS model) | | | Speech enhancement training material: (Vox2+ $Q_{Vox2+DNSMOS}$) | | |
| --- | --- | --- | --- | --- | --- | --- |
| | LCC | SRCC | MSE | SIG | BAK | OVRL |
| Noisy | - | - | - | **4.47** | 3.40 | 3.50 |
| CNN | **0.8542** | **0.7892** | $5.45 \times 10^{-5}$ | 4.33 | **4.02** | **3.73** |
| BLSTM | 0.7833 | 0.7498 | $7.54 \times 10^{-5}$ | 4.38 | 3.62 | 3.62 |
| CNN-BLSTM | 0.8513 | 0.7781 | $\mathbf{5.18 \times 10^{-5}}$ | 4.26 | 3.78 | 3.62 |

Although the three models can achieve similar quality prediction accuracy, the SE performances are quite different. If the quality predictor contains a BLSTM structure, the SE performance is generally worse than that without BLSTM. We argue that it is because the recurrent characteristics make the gradient hard to directly guide the front-end SE model. In summary, **a speech quality predictor with better accuracy may not necessarily be an appropriate teacher to guide another model**.

In Table 2, compared to the results shown in Table 1, $Q_{Vox2+DNSMOS}$ can guide the training of SE with a much higher BAK but a lower SIG. Again, if the quality predictor contains a BLSTM structure, its SE performance is much worse than CNN. Learning curves and enhanced spectrograms of these three model structures can be found in Appendix A. We find that if the predictor contains BLSTM structure, some unnatural band-like artifacts are generated. Hence in the following experiments, a CNN will be used for quality prediction.

### 4.2.2 COMPARISON BETWEEN DEEPDREAM-LIKE TRAINING AND GAN-LIKE TRAINING

In the previous section (DeepDream-like training), we showed that $Q_{Vox2+DNSMOS}$ can also guide the learning of a SE model. Therefore, we want to compare it with the discriminator ($D$) in GAN-like training. The main difference between these two models is that $Q_{Vox2+DNSMOS}$ is first pre-trained with a large amount of training data and once the training converges, its weights are permanently fixed. On the other hand, $D$ is randomly-initialized, and its weights will be updated along with the GAN training. Fig. 1 shows the learning curves with different metrics (DNSMOS and DNSMOS P.835) of these two training methods on the validation set of Vox2. Note that the start point (iteration 0) is the corresponding score of noisy speech. We can first observe that for $Q_{Vox2+DNSMOS}$, it reaches its maximum score (except for the SIG case) only within a few iterations and the scores start to decrease after that. In addition, the behavior of the two models is very similar to each other for the first few iterations. This finding seems to align with the observation made in (Fu et al., 2019a), that for fixed $Q_{Vox2+DNSMOS}$, only the gradient from the first few iterations can effectively guide the SE model. We argue that this is related to the generation scheme of adversarial examples (Yuan et al., 2019) and the model is fooled (estimated quality scores still increase but true scores decrease) (Nguyen et al., 2015). Adversarial training (Tramèr et al., 2017) is an effective way to prevent the model from being fooled, this explains why the SE model guided by $D$ can keep improving. Because GAN-like training performs better, the following experiments will be based on this training framework.

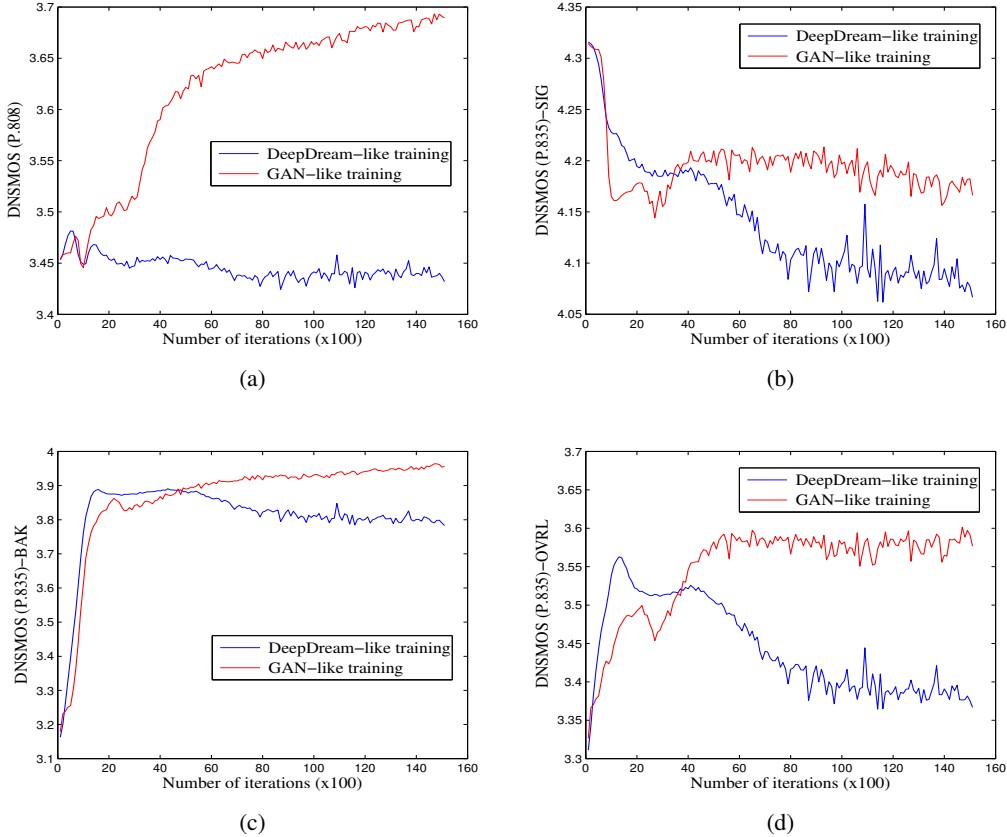

Figure 1: Learning curves of DeepDream-like training and GAN-like training on the validation set of Vox2 with different metrics (a) DNSMOS (P.808), (b) SIG, (c) BAK and (d) OVRL.

### 4.2.3 SPEECH ENHANCEMENT RESULTS ON THE VOX2 AND 3RD DNS TEST SET

In this section, we compare different SE model training methods on the Vox2 and 3rd DNS test sets. In addition to CNN-BLSTM, DPT-FSNet (Dang et al., 2022) is also employed to serve as state-of-the-art (SOTA) of conventional supervised training. They are trained on the VoiceBank-DEMAND data set and can reach PESQ scores 2.80 and 3.33 on the VBD test set, respectively. In Table 3, it can be observed that for the CNN-BLSTM, GAN-like training can outperform conventional supervised training in terms of BAK, and OVRL which implies it has better noise removal ability. On the other hand, its SIG is worse than the baseline, perhaps because it never sees a corresponding clean reference during training. The semi-supervised training uses a supervised model as initial SE model and then applies the same training framework as GAN-like training. For CNN-BLSTM, the results from semi-supervised can outperform the supervised baseline by a large margin, which verifies real noisy training data can further improve the SE performance (learning curves can be found at Appendix B). Although DPT-FSNet can already reach SOTA performance on the VBD test set, applying semi-supervised learning can also boost its scores.

For the 3rd DNS test set, we also compare our methods with others that can use real noisy speech during SE training. The results of MixIT (Wisdom et al., 2020), and Modifed MixIT (Trinh & Braun, 2021) come from (Trinh & Braun, 2021) and Convolutional Recurrent U-net for Speech Enhancement (CRUSE) (Braun et al., 2021) is chosen as SE model structure. Note that because the model structure and the training material used in (Trinh & Braun, 2021) are different from ours, their results are just for reference not for direct comparison. From Table 4, it can be observed that, compared to noisy, the improvement brought by MixIT is somewhat limited. On the other hand, as shown by Modified MixIT, applying the embedding from Wav2vec 2.0 during training can further improve the

results. Our methods basically follow the same trend as in the Vox2 test set, for CNN-BLSTM, the results from GAN-like training can outperform the supervised baseline. In addition, semi-supervised learning can also further boost the performance both for CNN-BLSTM and DPT-FSNet. Compared to GAN-like training, Robust Q (Section 3.2.3) generally has higher SIG but lower BAK. For OVRL, Robust Q performs better in the more mismatched condition (Table 4), we argue it is because there is no SE model in this training method, hence generalization issue of SE model doesn't exist. (please see Appendix C and D for spectrogram comparision, and inference process of Robust Q, respectively.)

Table 3: Comparison of different SE training on the Vox2 test set. For training material, (a) represents clean (VoiceBank) + noise (DEMAND), (b) represents noisy (Vox2) + DNSMOS model.

| SE model | Training | Training material | DNSMOS | SIG | BAK | OVRL |
|---|---|---|---|---|---|---|
| Noisy | - | - | 3.566 | 4.47 | 3.40 | 3.50 |
| Wiener | - | - | 3.484 | 4.00 | 3.94 | 3.59 |
| - | Robust Q | (b) | 3.761 | 4 35 | 4.05 | 3.78 |
| CNN-BLSTM | Supervised | (a) | 3.804 | 4.34 | 4.16 | 3.80 |
| | GAN-like training | (b) | 3.795 | 4.23 | 4.34 | 3.89 |
| | Semi-supervised | (a) + (b) | 3.922 | 4.33 | 4.40 | 3.99 |
| DPT-FSNet | Supervised | (a) | 4.000 | **4.49** | 4.41 | 4.02 |
| | Semi-supervised | (a) + (b) | **4.050** | **4.49** | **4.54** | **4.13** |

Table 4: Comparison of different SE training on the DNS3 test set. For training material, (a) represents noisy (Vox2) + noise (DNS), (b) represents clean (VoiceBank) + noise (DEMAND), and (c) represents noisy (Vox2) + DNSMOS model.

| model | Training | Training material | DNSMOS | SIG | BAK | OVRL |
|---|---|---|---|---|---|---|
| Noisy | - | - | 2.934 | 3.87 | 3.05 | 3.11 |
| Wiener | - | - | 2.928 | 3.71 | 3.25 | 3.12 |
| CRUSE | MixIT | (a) | - | 3.80 | 3.28 | 3.16 |
| | Modifed MixIT | (a) + Wav2vec 2.0 | - | 3.69 | 4.00 | 3.29 |
| - | Robust Q | (c) | 3.131 | 3.84 | 3.51 | 3.28 |
| CNN-BLSTM | Supervised | (b) | 3.166 | 3.66 | 3.77 | 3.18 |
| | GAN-like training | (c) | 3.189 | 3.66 | 3.91 | 3.25 |
| | Semi-supervised | (b) + (c) | 3.312 | 3.65 | 3.98 | 3.28 |
| DPT-FSNet | Supervised | (b) | 3.339 | 3.80 | 4.07 | 3.37 |
| | Semi-supervised | (b) + (c) | **3.457** | **3.88** | **4.17** | **3.49** |

### 4.2.4 RESULTS OF LISTENING TEST

To evaluate the subjective opinion of the enhanced speech, we conducted listening tests to compare the proposed semi-supervised training methods with supervised baselines and noisy speech. Experimental results show that our proposed training method can outperform conventional supervised training especially under mismatch conditions (please see Appendix E for more detail).

## 5 CONCLUSION

In this study, we proposed a novel SE training method that can leverage real noisy speech. Speech quality prediction and speech enhancement is connected and deeply discussed. An adversarially robust quality predictor itself can directly be used for speech enhancement without the need to train a separate SE model. On the other hand, although some model architectures may have better quality prediction accuracy, they may not necessarily be an appropriate teacher to guide a SE model. Under the same SE model structure, our proposed training method can outperform conventional supervised training. In addition, when combining these two training methods, the results of semi-supervised learning show further improvements both objectively and subjectively.

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

# Appendix

## A  SPEECH ENHANCEMENT RESULTS USING DIFFERENT QUALITY PREDICTOR STRUCTURES

In sec 4.2.1, we have shown different model structures of quality predictor may result in very different SE performance. In this section, we further present the learning curves for the DeepDream-like SE model training with the three structures (i.e., CNN, BLSTM, and CNN-BLSTM) of $Q_{Vox2+DNSMOS}$ in Fig. 2. It can be observed that the curves are very different to each other, and except for SIG, CNN performs the best in other three metrics. Note that the results reported in Table 2 are based on the model that has maximum OVRL score on the validation set. In Fig. 3, the enhanced spectrograms by different quality predictors are also shown. From the figure, we find that if the predictor contains BLSTM structure, some unnatural band-like artifacts are generated. This also aligns with the conclusion made in section 4.2.1 that recurrent structure may not be suitable to guide the SE model training.

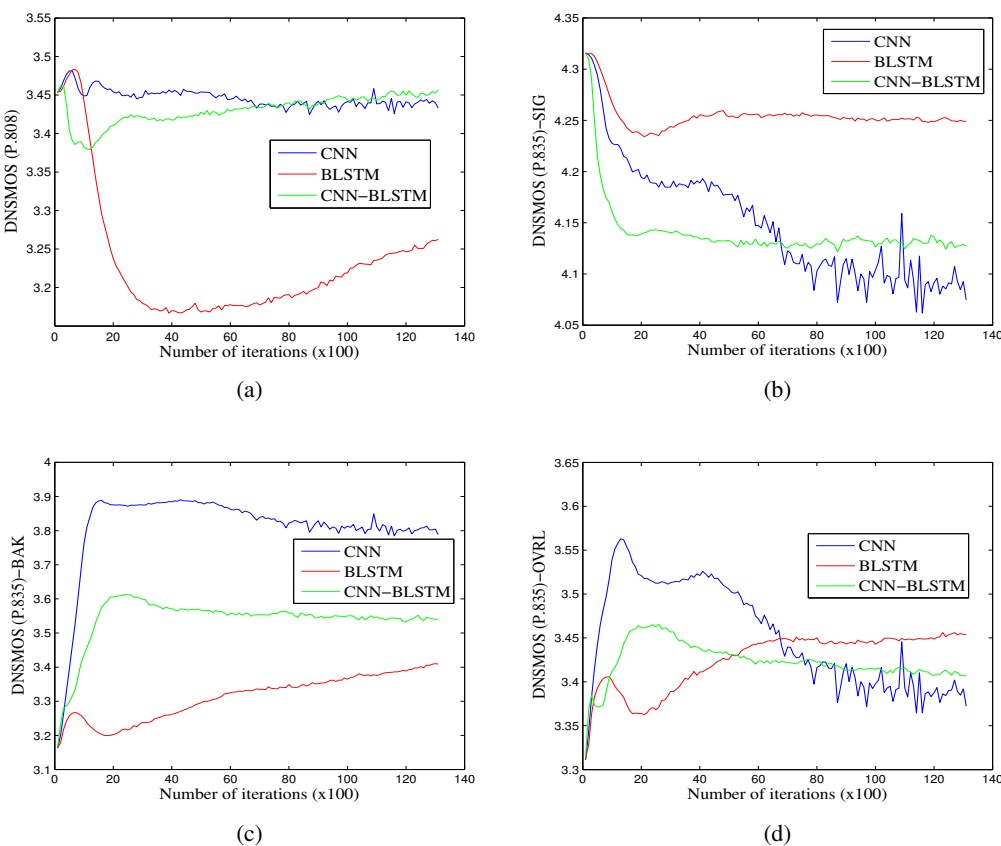

Figure 2: Learning curves of DeepDream-like SE model training using different quality predictor structures on the validation set of Vox2 (a) DNSMOS (P.808), (b) SIG, (c) BAK and (d) OVRL.

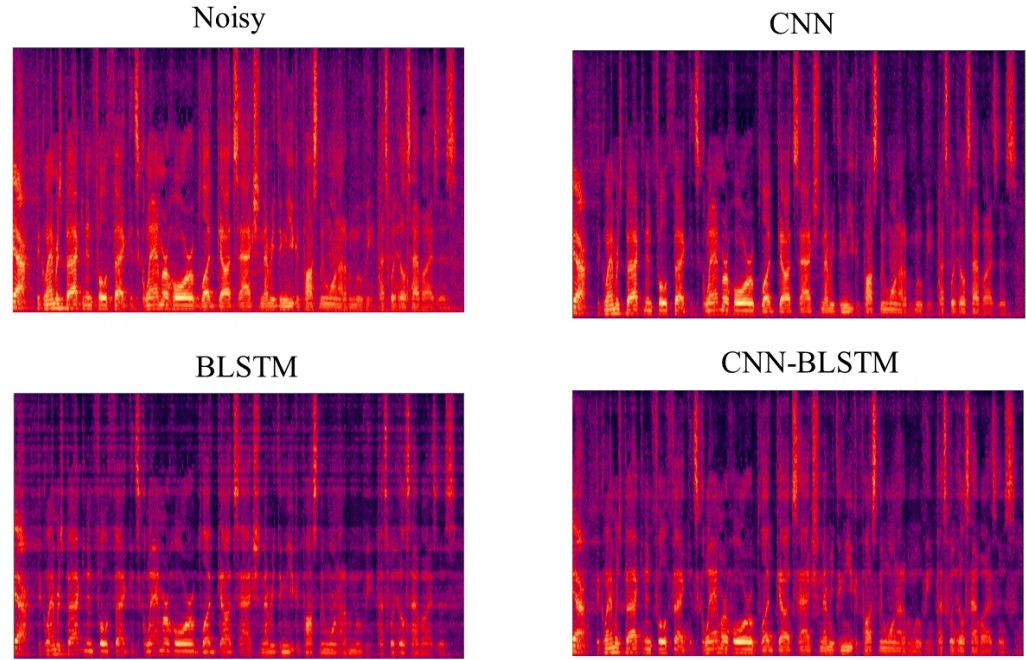

Figure 3: Enhanced spectrograms comparison by DeepDream-like training using different quality predictor structures.

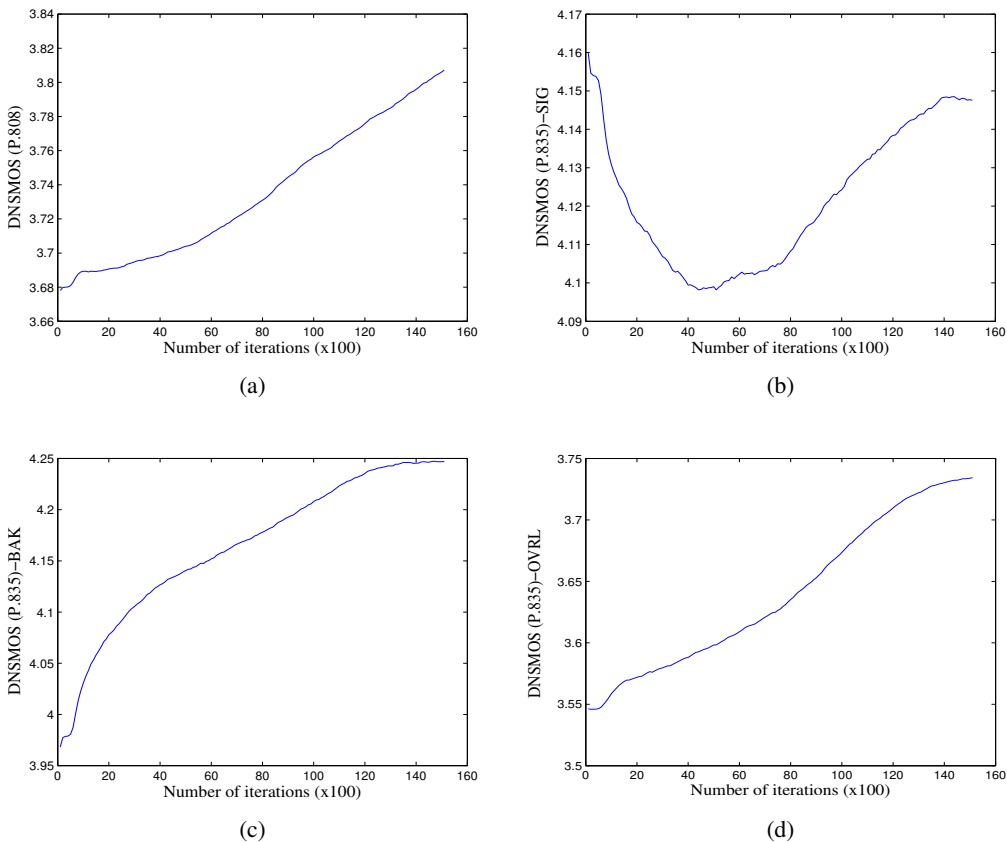

Figure 4: Learning curves of semi-supervised training using CNN-BLSTM as SE model on the validation set of Vox2 (a) DNSMOS (P.808), (b) SIG, (c) BAK and (d) OVRL. Note that the start point (iteration 0) shows the corresponding score of **enhanced speech** by supervised training baseline.

## B    LEARNING CURVES OF SEMI-SUPERVISED TRAINING

In Fig. 4, the learning curves of semi-supervised training (using CNN-BLSTM as SE model) are presented. In our semi-supervised training. we use the supervised trained model as our initial model, and then the proposed GAN-like training algorithm is applied. From this figure, it can be observed that except for SIG, the other three scores can gradually improve when real noisy speeches are used in SE model training.

## C    SPECTROGRAM COMPARISON BETWEEN NOISY AND ENHANCED ONES

In Fig 5, we show an example of spectrogram comparison between noisy and enhanced ones from different SE methods. In this figure, we show that GAN-like training can successfully remove the noise without the need of any synthethic training data. In the case of DPT-FSNet, semi-supervised training can remove more noise (as highlighted in the rectangle region) and keep speech components (in the circle regions) compared to the supervised baseline.

## D    ESTIMATED NOISE AND ENHANCED SPEECH BY ROBUST Q

In Section 3.2.3, we propose a SE method based on a robust quality predictor and gradient back-propagation to revise the input noisy speech. In Fig. 6, we show an example of how Robust Q

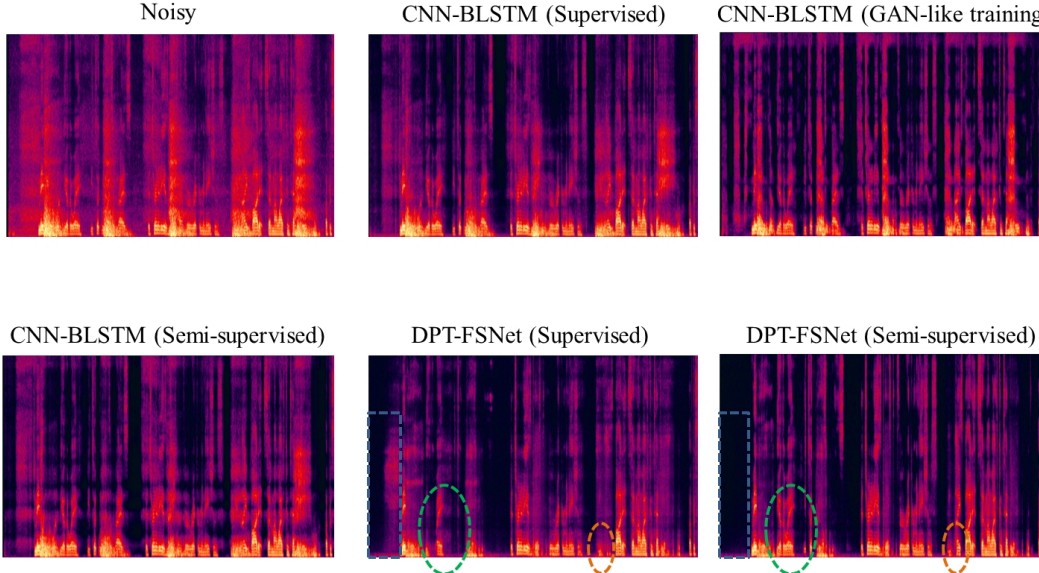

Figure 5: Spectrogram comparison (an example from the DNS3 test set) between noisy and enhanced spectrograms from different SE methods.

estimate noise spectrogram, **noise** and enhanced spectrogram, $\mathbf{Y}$ under differ iteration number $I$ in Algorithm 1. In general, the enhanced spectrogram can converge within 15 iterations.

## E    RESULTS OF LISTENING TEST

Because the blind test set in DNS1 (Reddy et al., 2020) contains noisy speech without reverberation (**noreverb**), noisy real recordings (**real**), and noisy reverberant speech (**reverb**), we believe this can cover different acoustic conditions for a subjective listening test. Specifically, this set comprised of 600 clips (300 synthetic and 300 real recordings). The real recordings data is collected using Amazon Mechanical Turk (MTurk). The MTurk participants captured their speech in a variety of noisy acoustic conditions and recording devices (headphones and speakerphones). The objective scores using different SE methods for **real**, **noreverb**, and **reverb** are first shown in Table 5, 6, 7, respectively. Interestingly, as shown in Table 6, when the testing condition most matches the supervised training condition (synthetic and noreverb), the gain brought by our proposed method is most limited. On the other hand, as shown in Table 7, when the testing condition most mismatches the supervised training condition, the performance gain of the proposed training method is most obvious. In addition, although DPT-FSNet usually performs better than CNN-BLSTM, its performance gets serious degradation when tested under **reverb** conditions. We argue that it is because DPT-FSNet has superior mapping ability only in the matched acoustic conditions.

To evaluate the subjective opinion of the enhanced speech, we conducted listening tests to compare the proposed semi-supervised training methods with supervised baselines and noisy speech. For each acoustic conditions (**real**, **noreverb**, and **reverb**), 7 samples were randomly selected from the test set; therefore, there were a total of $7 \times 5$ (different enhancement methods and noisy) $\times 3$ (acoustic conditions) = 105 utterances that each listener had to take. For each signal, the listener rated the speech quality ($SIG_{sub}$), background noise quality ($BAK_{sub}$), and the overall quality ($OVRL_{sub}$) follows ITU-T P.835. 12 listeners participated in the study. Table 8, 9, 10 show the listening test results for **real**, **noreverb**, and **reverb**, respectively. In general, for DPT-FSNet, comparing supervised and proposed semi-supervised training method, we can observe that under a match condition (**noreverb**), the OVRL score performs very similarly: 4.02 (Supervised) and 4.01 (proposed Semi-supervised). On the other hand, from Tables 8 and 10, in the mismatch condition (**real**, and **reverb**), the OVRL score can improve from 3.55 (Supervised) to 3.64 (proposed Semi-supervised) and from 1.51 (Supervised) to 1.73 (proposed Semi-supervised), respectively.

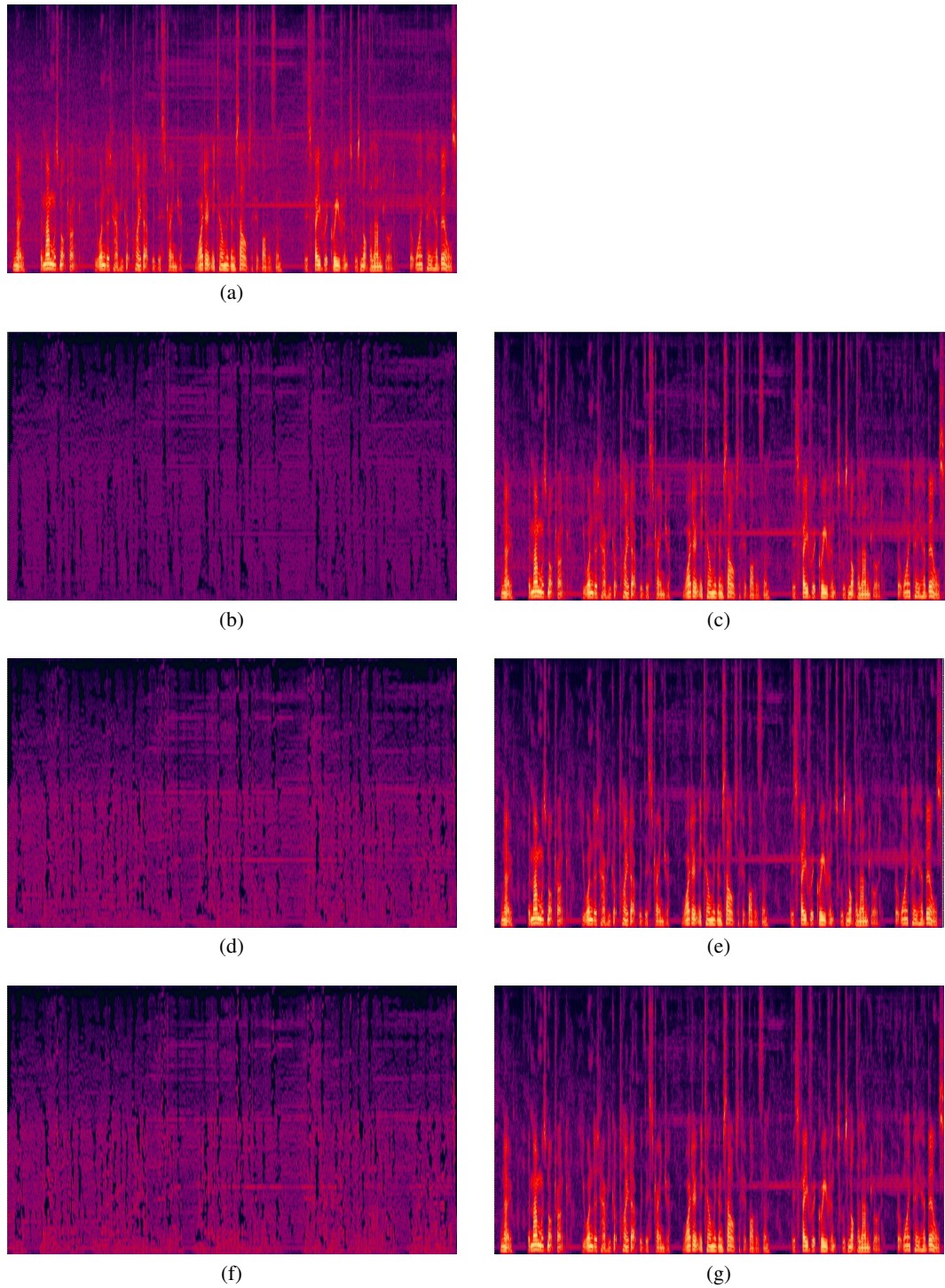

Figure 6: Estimated noise and enhanced spectrogram by Robust Q with different iteration number $I$. (a) noisy speech, (b), (d) and (f) estimated noise at iteration 1, 5 and 10, respectively. (c), (e), and (g) enhanced speech at iteration 1, 5 and 10, respectively.

Table 5: Comparison of different SE models on the DNS1 **real** test set. For training material, (a) represents clean (VoiceBank) + noise (DEMAND), (b) represents noisy (Vox2) + DNSMOS model.

| model | Training | Training material | DNSMOS | SIG | BAK | OVRL |
|---|---|---|---|---|---|---|
| Noisy | - | - | 3.086 | **4.18** | 2.93 | 3.25 |
| - | Robust Q | (b) | 3.297 | 4.01 | 3.56 | 3.37 |
| CNN-BLSTM | Supervised | (a) | 3.324 | 3.91 | 3.67 | 3.31 |
| | GAN-like training | (b) | 3.360 | 3.89 | 3.98 | 3.44 |
| | Semi-supervised | (a) + (b) | 3.471 | 3.89 | 3.99 | 3.47 |
| DPT-FSNet | Supervised | (a) | 3.518 | 4.06 | 3.93 | 3.49 |
| | Semi-supervised | (a) + (b) | **3.593** | 4.07 | **4.12** | **3.59** |

Table 6: Comparison of different SE models on the DNS1 **noreverb** test set. For training material, (a) represents clean (VoiceBank) + noise (DEMAND), (b) represents noisy (Vox2) + DNSMOS model.

| model | Training | Training material | DNSMOS | SIG | BAK | OVRL |
|---|---|---|---|---|---|---|
| Noisy | - | - | 3.276 | **4.49** | 3.43 | 3.55 |
| - | Robust Q | (b) | 3.494 | 4.31 | 3.90 | 3.64 |
| CNN-BLSTM | Supervised | (a) | 3.678 | 4.37 | 4.15 | 3.83 |
| | GAN-like training | (b) | 3.646 | 4.15 | 4.27 | 3.75 |
| | Semi-supervised | (a) + (b) | 3.811 | 4.26 | 4.34 | 3.88 |
| DPT-FSNet | Supervised | (a) | 3.940 | 4.42 | 4.38 | 3.95 |
| | Semi-supervised | (a) + (b) | **3.952** | 4.35 | **4.48** | **4.01** |

Table 7: Comparison of different SE models on the DNS1 **reverb** test set. For training material, (a) represents clean (VoiceBank) + noise (DEMAND), (b) represents noisy (Vox2) + DNSMOS model.

| model | Training | Training material | DNSMOS | SIG | BAK | OVRL |
|---|---|---|---|---|---|---|
| Noisy | - | - | 2.830 | **3.89** | 2.36 | 2.87 |
| - | Robust Q | (b) | 3.149 | 3.88 | 3.23 | **3.19** |
| CNN-BLSTM | Supervised | (a) | 3.151 | 3.67 | 3.49 | 3.05 |
| | GAN-like training | (b) | 3.239 | 3.70 | **3.86** | 3.16 |
| | Semi-supervised | (a) + (b) | **3.337** | 3.64 | 3.81 | 3.13 |
| DPT-FSNet | Supervised | (a) | 2.730 | 3.29 | 3.40 | 2.68 |
| | Semi-supervised | (a) + (b) | 3.121 | 3.45 | 3.74 | 2.91 |

Table 8: **Listening test** results of different SE models on the DNS1 **real** test set. For training material, (a) represents clean (VoiceBank) + noise (DEMAND), (b) represents noisy (Vox2) + DNSMOS model.

| model | Training | Training material | $SIG_{sub}$ | $BAK_{sub}$ | $OVRL_{sub}$ |
|---|---|---|---|---|---|
| Noisy | - | - | **4.27** | 2.57 | 3.24 |
| CNN-BLSTM | Supervised | (a) | 4.10 | 3.03 | 3.41 |
| | Semi-supervised | (a) + (b) | 3.99 | 3.08 | 3.43 |
| DPT-FSNet | Supervised | (a) | 4.01 | **3.31** | 3.55 |
| | Semi-supervised | (a) + (b) | 4.12 | **3.31** | **3.64** |

Table 9: **Listening test** results of different SE models on the DNS1 **noreverb** test set. For training material, (a) represents clean (VoiceBank) + noise (DEMAND), (b) represents noisy (Vox2) + DNSMOS model.

| model | Training | Training material | $SIG_{sub}$ | $BAK_{sub}$ | $OVRL_{sub}$ |
|---|---|---|---|---|---|
| Noisy | - | - | 4.25 | 2.63 | 3.22 |
| CNN-BLSTM | Supervised | (a) | 4.17 | 3.42 | 3.67 |
|  | Semi-supervised | (a) + (b) | 4.19 | 3.52 | 3.71 |
| DPT-FSNet | Supervised | (a) | 4.29 | 3.88 | **4.02** |
|  | Semi-supervised | (a) + (b) | **4.36** | **4.01** | 4.01 |

Table 10: **Listening test** results of different SE models on the DNS1 **reverb** test set. For training material, (a) represents clean (VoiceBank) + noise (DEMAND), (b) represents noisy (Vox2) + DNSMOS model.

| model | Training | Training material | $SIG_{sub}$ | $BAK_{sub}$ | $OVRL_{sub}$ |
|---|---|---|---|---|---|
| Noisy | - | - | **3.80** | 2.69 | **3.24** |
| CNN-BLSTM | Supervised | (a) | 2.97 | 3.15 | 2.85 |
|  | Semi-supervised | (a) + (b) | 3.18 | **3.26** | 3.08 |
| DPT-FSNet | Supervised | (a) | 1.50 | 2.87 | 1.51 |
|  | Semi-supervised | (a) + (b) | 1.65 | 2.95 | 1.73 |

