# OpenReview forum: "NEW TRAINING FRAMEWORK FOR SPEECH ENHANCEMENT USING REAL NOISY SPEECH"
_ICLR.cc/2023/Conference — Submitted to ICLR 2023_

### Official Review · Reviewer_zuxz · 2022-10-25

**Confidence:** 3
**Correctness:** 4
**Technical Novelty And Significance:** 2
**Empirical Novelty And Significance:** 2
**Recommendation:** 6

**Clarity, Quality, Novelty And Reproducibility:**

The paper is written well and easy to follow.

The major ideas described in this paper are already described in the prior art.

(i) `3.2.1 DeepDream-like Training` - described in Fu et al. 2019

(ii) `3.2.2 GAN-like training` - described in Fu et al. 2022 (MetricGAN-U)

(iii) `3.2.3 Robust Quality Prediction model` - As described in the experimental section (4.2.3), this method is inferior to GAN-like training.

**Strength And Weaknesses:**

Strength:
(i) Paper clearly describes the use of quality predictor for speech enhancement.
(ii) Empirical results demonstrate the usefulness of the proposed GAN-like training for speech enhancement models.
(iii) Appendix contains additional studies which further strengthen the usefulness of GAN-like training.

Weakness:
(i) There are bigger datasets for speech enhancements such as DNS (https://www.microsoft.com/en-us/research/academic-program/deep-noise-suppression-challenge-icassp-2022/) which contains over 500 hours of clean speech. On the other hand, the Quality Predictor dataset (IUB) is around 45 hours.
- Although the GAN-like training is useful, it would be interesting to see how much they help with really powerful SE baselines.
(ii) Semi-supervised training = GAN-like training with a fully-supervised SE model.
- The reverse should also be tried -> GAN-like training followed by fully-supervised training.
(iii) Comparison with related methods such MetricGAN-U (Fu et al. 2022) are missing.

Questions:
(i) Any particular reason metrics like PESQ, etc. were not evaluated. (Line 4 of Section 4.2.3 has a passing remark on PESQ of a prior-art).

(ii) `Because the calculation of PESQ and training of NORESQA rely on two signal processing measures, Signal-to-Noise Ratio (SNR) and Scale-Invariant Signal to Distortion Ratio (SI-SDR) to compare the quality of the two inputs, synthetic data is needed to train the quality prediction model. However, in our proposed training method, it is not necessarily needed.`
- The paper proposes a training of quality prediction model using a real dataset, why wouldn't that be used? What did I miss?

**Summary Of The Paper:**

This paper describes the method for training the speech enhancements model using only real noisy speech (as opposed to the typical use of labeled clean and noisy speech). The paper describes the use of a Quality Predictor (Q) to guide the training of the speech enhancement model.

The main contributions of the paper are:
(i) GAN-like training (Bi-level optimization) to train the quality predictor and speech enhancement model jointly which improves over the baseline.
(ii) Use of adversarial robust quality predictor as a speech enhancement model.

**Summary Of The Review:**

The paper describes a training framework for speech enhancement using real noisy speech, it lacks enough novel ideas. The most important ideas described in this paper are already tried. I really appreciate the thoroughness of the experiment section and appendix.

---

### Official Review · Reviewer_6yTX · 2022-10-25

**Confidence:** 3
**Correctness:** 2
**Technical Novelty And Significance:** 4
**Empirical Novelty And Significance:** 2
**Recommendation:** 5

**Clarity, Quality, Novelty And Reproducibility:**

Clarity (see above as well):
* One opportunity to simplify the experiments would be to fix the unobtrusive speech quality prediction model. I think that the inclusion of the training of the quality predictor as a factor under investigation muddied the water, since sometimes a separate speech quality predictor was used to train the speech quality predictor under investigation. Separating the data and models used for quality prediction and speech enhancement would significantly streamline the message of the paper.

Reproducibility:
* The abstract states that code will be released after publication but it is not submitted as supplementary material.

**Strength And Weaknesses:**

Strengths:
* Important problem. There is a great deal of noisy speech out there, much more than the amount of clean speech. It is also much easier to acquire noisy speech that is matched to a particular application than to create it synthetically. Thus a speech enhancement system that could be trained from such noisy speech alone, without the need for a clean reference, would be very valuable.
* Novelty: While other papers have trained speech enhancement systems using reference-based speech quality or intelligibility predictors, to my knowledge this is the first paper using a reference-free predictor.

Weaknesses:
* Clarity. I find the paper very hard to follow. Specifically, as mentioned in my summary, many many variations of systems and experiments are compared and it is difficult to confirm the claims that the paper makes with regards to these results or follow the chain of reasoning through the entire set of experiments. I am very familiar with the field in general, but just understanding which version of which approach was trained on which data and what that meant for the assumptions or hypotheses that were being evaluated in each experiment was very difficult to follow.
* Proposed speech enhancement model is a bit out of date. The speech enhancement component of the system used a magnitude-only mask that was constrained to lie between 0 and 1, citing papers from 2018. This is ok, but since then, most approaches have shifted to approaches like TAS-Net or masking of complex signals. If the goal is to maximize quality, then it would be much more realistic to go back to the time domain from the enhancer before going into the quality estimator so that any phase artifacts or effects are properly accounted for (i.e., by showing up in the magnitude spectrogram after the resynthesis-reanalysis process).

**Summary Of The Paper:**

This paper describes a method to train a speech enhancement system using a non-intrusive speech quality estimator. Many combinations of approaches, architectures, datasets, and training schedules are compared and measured with many metrics. The proposed approach does not do as well as the supervised or semi-supervised DPT-FSNet model, but does improve overall quality as estimated either through listening tests or DNSMOS p.835 compared to the original noisy speech.

**Summary Of The Review:**

This paper approaches a significant problem with a potentially novel solution, but the clarity of the explanation makes it difficult to understand and to confirm.

---

### Official Review · Reviewer_kbKN · 2022-10-25

**Confidence:** 5
**Clarity, Quality, Novelty And Reproducibility:** See S&W section.
**Correctness:** 3
**Technical Novelty And Significance:** 2
**Empirical Novelty And Significance:** 2
**Recommendation:** 3

**Strength And Weaknesses:**

Strengths

- Training with light supervision such as quality scores is an interesting and important problem in ML.
- In general the problem and approaches taken are adequately explained.
- As discussed above, the results suggest that the presented SE approaches lead to gains.

Limitations

- The results are solely based on the subjective quality of enhanced speech on noisy test sets with no clean reference. Moreover, most of these quality assessments are model generated, i.e. based on DNSMOS P.835 QPM scores. It is encouraging that the AMT and QPM scores follow the same trends, but traditional metrics such as SNR gain on datasets with parallel clean data should also be included.
- As mentioned in the summary, their semi-supervised systems are trained on DNSMOS P.808 external QPM model scores, which, to the extent that P.808 and P.835 are similar by virtue of training data and otherwise, may seriously affect the integrity of the semi-supervised results.
- Following up on the previous points, it is very difficult to gauge the signficance of these gains in quality score without audio examples, and the lack of traditional metrics such as SNR gain makes this judgement even more difficult.
- In addition, the authors compare to only one recent SOTA approach for audio separation (MixIT, and Modified MixIT), which was not even formulated as an SE approach. This also makes the signficance of the results very difficult to assess.
- While the presented methods may be novel to the SE application, they are already well established ML techniques, making the novelty and signficance of the paper for a method-focused ML conference low.
- The grammar in the paper is slightly below par for an ICLR paper, which requires another revision.

Minor Corrections:

used to prevent enhanced magnitude spectrogram Y has T-F bins smaller than zero,
->used to prevent the enhanced magnitude spectrogram Y from having T-F bins smaller than zero,

To utilize both benefits of synthetic clean/noisy speech pairs (easy to learn) and real noisy speech (large amount),
->To realize the benefits of utilizing both synthetic clean/noisy speech pairs (supervised) and real noisy speech (plentiful) during training,

The quality predictor is then concatenated after a randomly-initialized SE model.
->The quality predictor is then applied to the output of a randomly-initialized SE model.

non-intrusive (no clean reference is needed)
->If you really want to use this terminology (it is counterintuitive), define it earlier in the paper.


**Summary Of The Paper:**

The authors consider the problem of training speech enhancement (SE) models using only noisy speech utterances an their associated quality assessments, which are either annotated, or predicted by a quality prediction model (QPM). Several related approaches are considered, including 1) QPM score based SE optimization with a fixed QPM 2) joint GAN-like training of the SE and QPM, and 3) Enhancement based on an adversarially optimized QPM, by gradient-based noise removal of it's input features.

The approaches are evaluated primarily based on noisy testsets with no estimated clean reference (Vox2 and  DNS3), based on the DNSMOS P.835 external QPM model's scores, indicating small gains over MixIT and Modified MixIT (Their semi-supervised systems are trained on DNSMOS P.808 external QPM model scores, which, to the extent that P.808 and  P.835 are similar by virtue of training data and otherwise, may seriously affect the integrity of the semi-supervised results) . The approaches are also evaluated on 600 utterances of DNS1 by AMT workers, with small gains, modest gains, and degradation in overall quality on "real", "noreverb", and "reverb" subsets. Notably, the QPM model scores (tables 5,6,7) and AMT scores (tables 8,9,10) suggest the same performance trends, despite significant differences in their absolute scores, giving some credibility to utilizing external QPM models for evaluation.

**Summary Of The Review:**

Overall I feel that audio demonstrations, results on parallel datasets with traditional metrics such as SNR gain, and additional comparisons with SOTA SE approaches are needed before the paper is ready for publication. I also feel that the paper is perhaps better suited for an application-focused conference as the ML novelty of the paper is low.

---

> ### Comment · Reviewer_kbKN · 2022-12-02
> **Rebuttal Response**
>
> Thank you to the reviewers for their response. However, my concerns remain, and so I have not modified my score, and I am only more confident in my assessment. The audio examples clearly demonstate that improving these scores doesn't mean much. I can understand what the speakers are saying before enhancement, and after enhancement substantial artifacts remain. Removing a quasi-stationary sound from about half of the clip, and retaining it in the other half is not useful. Entirely failing to remove a bird singing over an otherwise clean recording is not useful. Furthermore, in ICLR you can update your paper, and reviewers expect you to go ahead and fix problems that are identified by them, not issue promissory notes. The ML novelty of the paper is low, the manuscript requires substantial revision, and the significance of the work is not clear. This paper should be rejected.

---

### Official Review · Reviewer_Pxpx · 2022-10-28

**Confidence:** 4
**Correctness:** 4
**Technical Novelty And Significance:** 3
**Empirical Novelty And Significance:** 3
**Recommendation:** 8

**Clarity, Quality, Novelty And Reproducibility:**

The paper is written very clearly.
The overall quality of presentation and writing is good.
Reproducibility is not fully clear in particular when it comes to the last details of data and hyperparameters.
However, release of code as promised will make things clear, supposedly.

**Strength And Weaknesses:**

The method is simple, yet appears effective. It is well described and tested sufficiently.
Having more audio references would be a good addition.
Also, seeing more on ablation studies, comments on full reproducibility, additional comparative quantative result interpretetion and limitation considerations would make this an even stronger candidate.

**Summary Of The Paper:**

The authors suggest an approach towards directly exploiting originally noisy speech w/o the need for a clean reference.
The algorithm is presented and code will be released. A combination with "common" denoising by having a reference available is possible.

**Summary Of The Review:**

A good paper in terms of writing and quality.
What's more - the approach seems to be working and is rather a potential game changer in SE and beyond, as this could also be applied to other signal types in the noise.

---

### Decision · Program_Chairs · 2023-01-20

**Decision:**

Reject

**Justification For Why Not Higher Score:**

Issues with clarity of presentation, and limited novelty and performance.

**Justification For Why Not Lower Score:**

N/A

**Metareview: Summary, Strengths And Weaknesses:**

The authors propose a strategy for training with real (and synthetic) noisy speech, making use of pre-fixed or pre-trained or jointly trained speech-quality prediction models. The authors also show that you get better results when training the quality prediction model in a GAN-like fashion or via adversarial training.

There have been prior works that try to use noisy speech directly for enhancement in various fashions. Similarly, there are also approaches that use metric computation for enhancement. Novelty here is in the unique application of a quality prediction model for training enhancement models using real noisy speech.

Reviewers raised concerns around clarity of presentation, to a degree that it hindered good understanding of the paper. The authors tried to clarify this in their rebuttal, but issues remain.

There were also concerns raised about the metric used for evaluation. One of the evaluation metrics used by the authors (DNS P.835) is similar to the one that they use for training (DNS P.808). The authors argued that the metric, although similar, are different models. But the two metrics are indeed highly correlated, which may bias the results. The evaluations on noisy test sets with no clean reference make it harder to provide wider comparisons with supervised / SOTA strategies, which would have helped.

Finally, the overall gains across test conditions are modest. The authors shared audio examples, and the general opinion from the reviewers is that the quality of improvement is minor.

**Summary Of Ac-Reviewer Meeting:**

This paper was borderline. The main discussion points during the meeting:
- The paper needs more revision, and lacks clarity the way it is written now. The authors did not use the review comments to update the paper.
- In response to the reviews, the audio examples provided by the authors were not convincing enough. In a lot of cases, it was difficult to understand how enhancement was helping.
- The reviewers did agree that the authors are addressing a very important problem, and that the method could be interesting with more compelling results and validation.